# Appropriate Reference Genes for RT-qPCR Normalization in Various Organs of *Anemone flaccida* Fr. Schmidt at Different Growing Stages

**DOI:** 10.3390/genes12030459

**Published:** 2021-03-23

**Authors:** Zeying Zhao, Hanwen Zhou, Zhongnan Nie, Xuekui Wang, Biaobiao Luo, Zhijie Yi, Xinghua Li, Xuebo Hu, Tewu Yang

**Affiliations:** 1MOA Key Laboratory of Crop Ecophysiology and Farming System in the Middle Reaches of the Yangtze River, College of Plant Sciences and Technology, Huazhong Agricultural University, Wuhan 430070, China; zhaozeyingme@163.com (Z.Z.); zhouhanwen1990@126.com (H.Z.); wang-xuekui@mail.hzau.edu.cn (X.W.); luo_piaopiao@163.com (B.L.); yizhijie11@163.com (Z.Y.); lixh199015@163.com (X.L.); xuebohu@mail.hzau.edu.cn (X.H.); 2Northwest Institute of Eco-Environment and Resources, Chinese Academy of Sciences, Lanzhou 730030, China; 3University of Chinese Academy of Sciences, Beijing 100049, China; 4Department of Jobs: Precincts and Regions, Private Bag 105, Hamilton, VIC 3300, Australia; zhongnannie@gmail.com

**Keywords:** *Anemone flaccida* Fr. Schmidt, reference gene, data normalization, RT-qPCR, medicinal herb

## Abstract

*Anemone flaccida* Fr. Schmidt is a traditional medicinal herb in southwestern China and has multiple pharmacological effects on bruise injuries and rheumatoid arthritis (RA). A new drug with a good curative effect on RA has recently been developed from the extract of *A. flaccida* rhizomes, of which the main medicinal ingredients are triterpenoid saponins. Due to excessive exploitation, the wild population has been scarce and endangered in a few of its natural habitats and research on the cultivation of the plant commenced. Studies on the gene expressions related to the biosynthesis of triterpenoid saponins are not only helpful for understanding the effects of environmental factors on the medicinal ingredient accumulations but also necessary for monitoring the herb quality of the cultivated plants. Reverse transcription quantitative polymerase chain reaction (RT-qPCR) as a sensitive and powerful technique has been widely used to detect gene expression across tissues in plants at different stages; however, its accuracy and reliability depend largely on the reference gene selection. In this study, the expressions of 10 candidate reference genes were evaluated in various organs of the wild and cultivated plants at different stages, using the algorithms of geNorm, NormFinder and BestKeeper, respectively. The purpose of this study was to identify the suitable reference genes for RT-qPCR detection in *A. flaccida*. The results showed that two reference genes were sufficient for RT-qPCR data normalization in *A. flaccida*. *PUBQ* and *ETIF1a* can be used as suitable reference genes in most organs at various stages because of their expression stabilitywhereas the *PUBQ* and *EF1Α* genes were desirable in the rhizomes of the plant at the vegetative stage.

## 1. Introduction

*Anemone flaccida* Fr. Schmidt, a perennial herb in the Ranunculaceae family, distributes mainly in southwestern China and is commonly known as “Di Wu” in its original place [1]. Its rhizome has been used as a traditional national medicine to heal fractures for a long time [2]. Recently, a new drug extracted from its rhizomes has been developed by the Hubei University of Chinese Medicine and has shown good curative effects in the treatment of rheumatoid arthritis (RA) [3].

The habitats of *A*. *flaccida* plants are usually under woods in mountainous areas over 1500 m above sea level [4]. In natural habitats, the sprouting buds of the plant usually emerge in mid- to late- March from the rhizomes and then grow in spring when the weather becomes warmer. The plant enters flowering and seed production from mid- April to late- May. In late June, its aboveground parts begin senescence and die eventually. The plant grows relatively vigorously in spring compared with that in winter; however, its growth rate is rather slow because of cool temperatures in the natural high altitude habitat. The total growth period of the species in a year is usually about 90 days. Most seeds of the plant cannot germinate probably because they often shatter before full maturity [4]; thus, the plant depends mainly on its rhizome for population propagation. The low reproductive capacity, slow growth characteristics and narrow habitat [5] have led the wild populations of the plant to be scarce and endangered due to excessive exploitation in the past years. In order to meet the requirement for medicinal usage, studies have been undertaken to grow the plant under a more controlled environment and management. For instance, Xin [6] showed that the combined application of phosphate and potassium fertilizers could promote the growth and development of the cultivated plants.

Quality control is of great importance for medicinal herb plantations. Apart from the quantification of medicinal ingredients, the expression analysis of the key genes involved in the biosynthesis of pharmacological constituents is often performed into monitor the herb quality [7]. Moreover, gene expressions can provide a deep insight to understanding how cultivation techniques and environmental factors affect the herb quality, which is conducive to developing cultivation and management methods. The main medicinal ingredients are triterpenoid saponins in the rhizome of *A*. *flaccida* [2]. In 2016, Zhan et al. [8] performed a combined transcriptomic and proteomic analysis to identify key genes in the biosynthetic pathway of triterpenoid saponins in *A*. *flaccida*, which made it possible to monitor the quality of cultivated plants through gene analysis.

Reverse transcription quantitative polymerase chain reaction (RT-qPCR), with the characteristics of high efficiency, high sensitivity and good repeatability, has been widely used to quantify the expression levels of transcripts in various organisms at various developmental stages and under different conditions [9,10]. In RT-qPCR analysis, the appropriate internal reference gene(s) for data normalization is an important prerequisite to ensure the accuracy. Reference genes used in RT-qPCR analysis usually play important roles in maintaining the normal life of cells and are often referred to as housekeeping genes, such as *actin* (*ACT*) [11], *18S ribosomal RNA* (*18S rRNA*) [11], *Elongation Factor 1-α* (*EF1A*) [12], *beta tubulin* (*β-tubulin*) [13], *glyceraldehyde-3-phosphate dehydrogenase* (*GAPDH*) [13], *pyruvate kinaseII* (*PKII*) [14], *alpha-tubulin* (*α-tubulin*) [15], *28S ribosomal RNA* (*28SrRNA*) [16], *polyubiquitin* (*PUBQ*) [17], *Eukaryotic translation initiation Factor 1A* (*ETIF1a*) [18], *ubiquitin* [18] and *histone H2A* (*hh2a*) [19]. Most of these genes usually express stably in different organs and at various developmental stages. However, a few of these housekeeping genes have been shown to express variably across species and tissues or under diverse conditions [12]. Failure to use suitable reference gene(s) may deviate from the gene expression profile and lead to misinterpretation [20]. Therefore, the suitable reference genes specific to the species, organs and growth conditions should be validated.

The best reference genes are usually identified by Excel-based software statistical algorithms such as geNorm [21], NormFinder [22] and BestKeeper [23]. Such identifications have been done in model plants, grain crops and a few other commercial crops such as *Arabidopsis* [24], rice (*Oryza sativa* L.) [25] and *Zanthoxylum bungeanum* Maxim [26]. However, in medicinal plants, similar studies have only been performed in a few major species, such as *Anoectochilus roxburghii* [13], *Panax ginseng* [27] and *Tripterygium wilfordii* [28]. In this study, the expression stability of 10 commonly used reference genes that were selected from a comprehensive literature search [12,13,14,15,16,17,18,19] and based on the transcriptomic data [8] was evaluated in different organs at various developmental stages of *A. flaccida*. The purpose of this study was to identify the appropriate reference gene(s) for the normalization of gene expression studies in the medicinal quality of cultivated plants in *A. flaccida*.

## 2. Materials and Methods

A field experiment was conducted at the Chinese herbal medicine plantation base (30°8′31″ N, 108°55′35″ E, 1600 m above sea level) in Lichuan City, Hubei Province, China. Cutting rhizomes with a length of 2–3 cm were planted about 5 cm deep into the soil with a row spacing of 25 cm and plant distance of 12.5 cm on 26 September 2014. The density of the rhizome plantation was 320,000 plants/hm^2^. Prior to the planting of the rhizomes, 337.5 kg/hm^2^ of urea, 1000 kg/hm^2^ of calcium superphosphate and 625.5 kg/hm^2^ of potassium magnesium sulphate were applied into the soil as a basal fertilizer [6]. The cultivated plants were shaded by one layer of sunshade net with 10.4% of light transmittance to imitate the natural habitat of the plant. No irrigation and herbicides were applied during the entire growing period.

The statistical analysis of the expression stability was performed for all of the tested samples (total) and separately for each growth stage for the rhizome samples and for the samples of leaves of cultivated and wild plants. Rhizomes and leaves were sampled at the vegetative stage on 3 April and flowering stage on 10 May 2015, respectively (CVR and CVL: rhizomes and leaves of the cultivated plants, respectively, at the vegetative stage; CFR and CFL: rhizomes and leaves of the cultivated plants, respectively, at the flowering stage), Rhizomes were collected from withering plants on 11 July 2015 (CWR: rhizomes of the cultivated plants at the withering stage). Leaf and rhizome samples of *A. flaccida* were also collected from wild plants at Tonggubao (30°17′22″ N, 110°49′11″ E, 1800 m above sea level), a natural habitat of the plant, in Changyang county, Hubei Province, China on 6 April, 13 May and 14 July 2015, respectively (WVR and WVL: rhizomes and leaves of the wild plants, respectively, at the vegetative stage; WFR and WFL: rhizomes and leaves of the wild plants, respectively, at the flowering stage; WWR: rhizomes of the wild plants at withering stage). Following thorough cleaning by tap water, the samples were immediately put into liquid nitrogen and brought to the laboratory for analysis. All samples were collected in three biological replicates.

### 2.1. Total RNA Isolation and Quality Control

Frozen samples (2 g of leaves and 1 g of rhizomes) were ground with a pre-cooled mortar and pestle in liquid nitrogen. Total RNA was extracted from the leaves and rhizomes by Trizol Reagent (Invitrogen, Carlsbad, CA, USA) following the manufacturer’s instructions on a Heraguard ECO Clean Bench (Thermo Fisher Scientific, Waltham, MC, USA). The quality and quantity of RNA were checked by agarose gel electrophoresis and spectrophotometry (NanoDrop Technologies, Wilmington, DE, USA). RNA samples with A260/A280 between 1.8–2.2 were used for further RT-qPCR analysis.

### 2.2. Candidate Reference Gene Selections

Based on the transcriptome data sequences of *A. flaccida* [8], the potential homologues of 10 commonly used housekeeping genes were selected as candidate reference genes in this study: *EF1A* [12], *β-tubulin* [13], *GAPDH* [13], *PKII* [14], *α-tubulin* [15], *28SrRNA* [16], *PUBQ* [17], *ETIF1a* [18], *ubiquitin* [18] and *hh2a* [19].

### 2.3. Primer Design and Amplification Efficiency Test

Primers for the amplification of candidate reference genes were designed using Oligo7.0 software (V7.56 version) based on the data of Zhan et al. [8] with an annealing temperature between 58 °C and 62 °C, a primer length of 20–24 bp, GC content about 45%–55% and an amplicon length of 126–156 bp (Table 1, Appendix A). For each pair of primers, a standard curve was generated using five different cDNA concentrations (1, 5^−1^, 5^−2^, 5^−3^ and 5^−4^ fold dilutions with three replicates, respectively) (Appendix A). The cycle threshold (Ct) values were automatically determined for each reaction using CFX-1000 PCR apparatus (Analytik Jena AG company, Berlin, Germany). The amplification efficiency (E) for each gene was determined with the slope of a linear regression model [12] using the Ct values and the following equation: $E=10^{-1/slpoe}$.

### 2.4. Reverse Transcription Quantitative Polymerase Chain Reaction (RT-qPCR)

The first strand cDNA was synthesized with 1 μg of high-quality total RNA and 1 μL of oligo dT primers (50 μmol/L) for each sample according to the Thermo RevertAid First Strand cDNA Synthesis Kit (Thermo Fisher Scientific, USA). The RT-qPCR assays were carried out with a CFX-1000 PCR apparatus and SYBR Premix Ex TaqII (Wuhan Khayal Bio-Technology Company, Wuhan, China) to test the transcription variability of the 10 candidate reference genes across the samples. The total reaction volume was 20 μL, containing 1 μL of diluted cDNA, 10 μL of 2× Ultra SYBR Mixture, 0.4 μL of each primer (10 μmol/L) and 8.2 μL of RNase-free water. The three steps of RT-qPCR program began with 95 °C for 10 min, 45 cycles of 95 °C for 15 s and 60 °C for 15 s, followed by 72 °C for 10 min. The dissociation curve was obtained by heating the amplicon from 56 °C to 95 °C. Each RT-qPCR reaction was carried out with three technical replicates and a template-free control.

### 2.5. Statistical Analysis on Gene Expression Stability

The expression stability of the 10 candidate genes was evaluated by Ct value [29] and three different algorithms: geNorm (http://medgen.ugent.be/jvdesomp/genorm/, accessed on 20 July 2018), NormFinder (http://www.mdl.dk/publicationsnormfinder.html, accessed on 20 July 2018) and BestKeeper (http://www.genequantification.de/bestkeeper.html, accessed on 20 July 2018). The geNorm algorithm calculates the expression stability (defined as the parameter M) of each candidate reference gene based on its expression level in a set of given samples. The gene with the lower M value indicates a higher expression stability [15,21]. The geNorm algorithm selects the optimal number of required reference genes by comparing the pair-wise variation (Vn/Vn+1) values between consecutively ranked genes. The cut-off values are usually recommended to be a default value of 0.15 [21,30]. Similarto the geNorm algorithm, the NormFinder algorithm uses a model-based approach to determine the expression stability of reference genes based on intra- and inter- group variations. Herein, the gene with the lowest mean expression stability value is considered to be the most stably expressed [23,31]. BestKeeper evaluates and ranks the expression stability of candidate reference genes by calculating the standard deviation (SD) and the coefficient of variation (CV) based on the Ct values. The most stably expressed reference gene presents the lowest standard deviation (CV ± SD) [24,32]. Following RT-qPCR data collection, amplification cycles (Ct values) were converted to relative quantities using the equation $E^{-\Delta \mathrm{Ct}}\;$(ΔCt = the corresponding Ct value − minimum Ct). The relative quantities were then used for the gene expression stability evaluation by geNorm, NormFinder and BestKeeper analyses, respectively, according to the manufacturer’s instructions.

## 3. Results

### 3.1. Validating Expression Levels of Candidate Reference Genes

The specificity of the primers used for the candidate reference genes were verified by reverse transcription PCR. There were no primer dimers and nonspecific amplification as checked by agarose gel electrophoresis (Appendix A). There were great variations in the transcript levels of the 10 candidate reference genes in the different samples of *A. flaccida* (Figure 1). The mean Ct values of the genes ranged from 22.82 to 28.06 across all samples. Among them, *PKII* had the lowest mean Ct value and *β*-*tubulin* had the highest mean Ct indicating the highest expression level for *PKII* but the lowest for *β*-*tubulin*. The expression level of *PUBQ* presented the least variation (CV = 0.2%), while that of *GAPDH* presented the most variation (CV = 5.78%) across all samples (Figure 1).

The relative quantitative expression of each gene in different organs at various stages is presented as the percentage of the aggregated reference transcript pool in Figure 2. The proportions of *EF1A*, *PUBQ*, *ETIF1a* and *β*-*tubulin* transcripts were relatively consistent while, those of *α*-*tubulin*, *28S rRNA*, *PKII*, *hh2a*, *GAPDH* and *ubiquitin* transcripts were more variable across the samples. The proportions of the gene transcripts were comparable in the same organs between wild and cultivated plants at the same stage, except for those of *PKII* and *28S rRNA* in the rhizomes at the vegetative stage, *ubiquitin* and *GAPDH* in the leaves at the vegetative stage, *GAPDH*, *PKII*, *hh2a* and *α-tubulin* in the rhizomes and leaves at the flowering stage, *ubiquitin* in the leaves at the flowering stages and *PKII* and *hh2a* in the rhizomes at the withering stage (Figure 2).

### 3.2. Expression Stability of Candidate Reference Genes

#### 3.2.1. geNorm Analysis

The tested reference genes by geNorm showed relatively stable expressions with M values less than 1.5 (Figure 3). The expressions of *PUBQ* and *EF1A* were the most stable while those of *28S rRNA* and *PKII* were the least stable across all samples (Figure 3a). At the vegetative stage, *PUBQ* and *EF1A* also kept the most stable expression in the rhizomes as well as *PUBQ* and *hh2a* in the leaves while *PKII* and *28S rRNA* showed the least stable expression in the rhizomes and *β-tubulin* and *ubiquitin* in the leaves (Figure 3b,c). At the flowering stage, the expressions of *ETIF1a* and *EF1A* were the most stable in the rhizomes as well as those of *ETIF1a* and *PUBQ* in the leaves (Figure 3d,e). The least stably expressed genes were *PKII* in the rhizomes and *GAPDH* in the leaves at the flowering stage. At the withering stage, *ETIF1a* and *EF1A* were the most stably expressed genes but *PKII* was the least stably expressed gene in the rhizomes (Figure 3f). Most V3/4 values (VR: 0.005; VL: 0.033; FR: 0.017; FL: 0.019) were larger than the V2/3 values (VR: 0.003; VL: 0.007; FR: 0.01; FL: 0.018) and all the of V2/3 values were below 0.15 (Figure 4). This indicated that two reference genes could be sufficient for normalizing gene expressions in *A*. *flaccida* samples. The addition of a third reference gene did not significantly increase the statistical reliability of the calculation.

#### 3.2.2. NormFinder Analysis

The expression stability of 10 candidate reference genes resulted from NormFinder analysis demonstrated that *PUBQ* was the most stably expressed gene in all samplesexcept for the rhizomes at the vegetative and withering stages in which *EF1A* was ranked as the most stably expressed gene (Table 2).

#### 3.2.3. BestKeeper Analysis

The results obtained by the BestKeeper analysis showed that all of the SD values of Ct of the 10 candidate reference genes were smaller than 1, and the expression of *PUBQ* was the most stable in total samples and in the rhizomes and leaves at the vegetative stage while *ETIF1a* was ranked as the most stable gene in the other samples (Table 3).

## 4. Discussion

The selection of suitable reference gene(s) is crucial for obtaining reliable and reproducible results of target gene expressions in different experimental conditions by RT-qPCR analysis [30,31]. An ideal reference gene should be relatively consistent in expression across all organs and experimental conditions [27]. In this study, *PUBQ* showed a stable expression in most cases for *A. flaccida*, which was in line with the previous findings in *Cicer arietinum* [17]. *GAPDH* has been recommended as a suitable reference gene in sugarcane (*Saccharum* sp.) [33]; however, its expression was not so stable for *A. flaccida* in this study. Similar results were also obtained in *Miscanthus lutarioriparia* [30]. These suggested that reference genes need to be verified under certain experimental conditions and among various species or organs [28,34].

The results of reference gene selections using RT-qPCR are also affected by reagents and analysis software. Three analytical software platforms and high-quality reagents were used to ensure the reliability of results in this study. geNorm, NormFinder and BestKeeper have been widely used to analyze the stability of gene expression and identify the best reference gene(s) for data normalization in RT-qPCR studies. In this study, *PUBQ* was ranked as the most stably expressed gene in all of the tested samples by the three methods. However, *EF1A* was ranked as the first and *PUBQ* as the second most stable gene in the rhizomes at the vegetative stage by The NormFinder. NormFinder algorithm uses a model-based approach to determine the expression stability of reference genes based on intra- and inter- group variation. BestKeeper evaluates and ranks the expression stability of candidate reference genes by calculating the standard deviation (SD) and the coefficient of variation (CV) based on the Ct values. The most stably expressed reference gene presents the lowest standard deviation (CV *±* SD) [23]. By BestKeeper, *ETIF1a* was ranked as the first most stable gene and *PUBQ* the second in different organs at the flowering and withering stages in this study although the *PUBQ* expression was the most stable at the vegetative stage.

The geNorm algorithm calculates the expression stability (defined as the parameter M) of each candidate reference gene based on its expression level in a set of given samples. The gene with the lower M value (M < 1.5) indicates a higher expression stability [21]. In the evaluation by geNorm, *ETIF1a* was deemed to be the most stable gene in all specific organs at the flowering and withering stages while *EF1A* was the most stably expressed gene in the rhizomes at various stages. Zhan et al. also used *EF1A* as a reference gene to quantify the expression levels of transcript in leaves and rhizomes of *A. flaccida* at the flowering stage [8]. *PUBQ* was ranked as the second and third most stably expressed gene in the rhizomes at the withering and vegetative stage, respectively, although it was ranked as the first most stably expressed gene in the leaves at the vegetative and flowering stages in this study. These confirmed that different methods may give different results because of their different strategies and calculation algorithms [29,35]. geNorm is known to be based on a more effective and feasible algorithm for ensuring the optimal stability of reference genes and its results have been widely accepted by many researchers [21]. NormFinder and BestKeeper are best applied to assess the quality of the gene rankings obtained by geNorm [31]. Moreover, the geNorm algorithm can yield the optimal number of required reference genes by comparing the pair-wise variation (Vn/Vn+1) values between consecutively ranked genes. The cut-off values are usually recommended to be a default value of 0.15 [30].

We combined the results from the three methods together in this study, which is shown in Appendix A. Based on the result by geNorm that two reference genes were sufficient for RT-qPCR data normalization in different organs at various stages, *PUBQ* and *ETIF1a* were the most suitable reference genes for the total sample and specific organs in *A. flaccida* at most stages. However, *EF1A* was more desirable as a reference gene than *ETIF1a* in rhizomes at the vegetative stage. Moreover, *EF1A* in combination with *PUBQ* could also be used as a reference gene in the rhizomes at the withering stage (Appendix A). In addition, the expression of *PUBQ* was more stable at the vegetative stage than at the other stages whereas the expression of *EF1A* was more stable in the rhizomes than in the leaves. These also confirmed that the expressions of a few housekeeping genes vary in spatial, temporal and environmental-dependent patterns in plants.

## 5. Conclusions

The genes *PUBQ* and *ETIF1a* had a stable expression in most organs at different developmental stages of *A. flaccida*. Therefore, they can be used as reference genes for the data normalization in an RT-qPCR analysis of this plant. *PUBQ* and *EF1A* were more desirable as reference genes in rhizomes at the vegetative stage and could also be used at the withering stage. This study laid a foundation for understanding the effects of cultivation methods on the biosynthesis of triterpenoid saponins in *A*. *flaccida* for future gene expression analysis, which will be beneficial for the development of artificial plantations of this medicinal herb.

## Figures and Tables

**Figure 1 genes-12-00459-f001:**
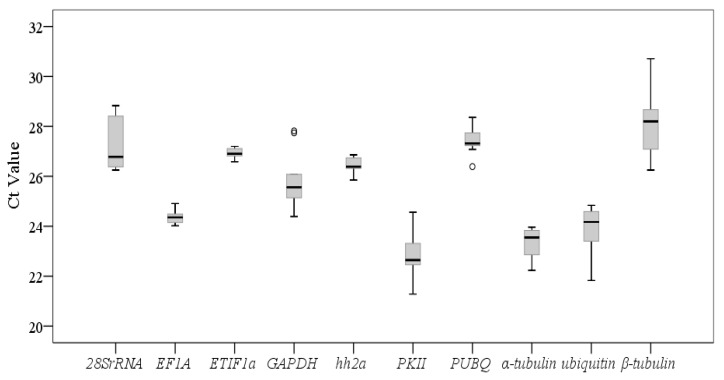
Expression levels of 10 candidate reference genes in different samples. Lines across the boxes depict the medians. Boxes indicate the interquartile range. Whiskers represent 95% confidence intervals and black circles represent the deviated values.

**Figure 2 genes-12-00459-f002:**
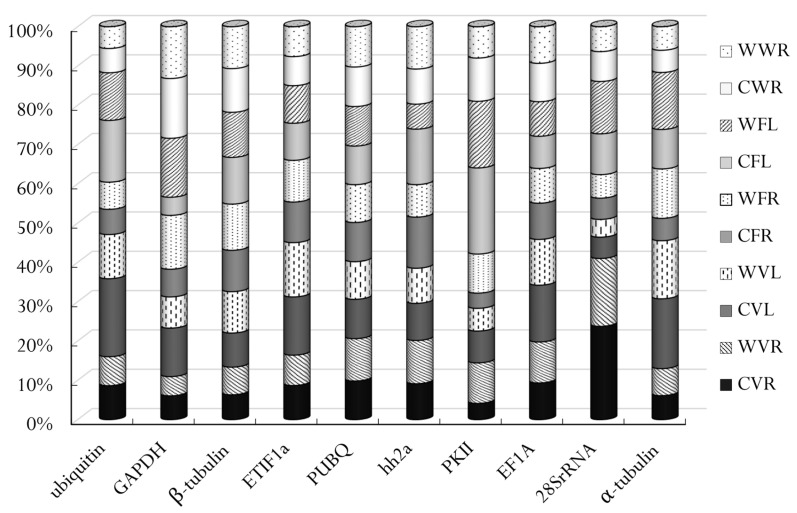
Relative expression levels of 10 candidate reference genes in different samples. CVR: rhizomes of the cultivated plants at the vegetative stage; WVR: rhizomes of the wild plants at the vegetative stage; CVL: leaves of the cultivated plants at the vegetative stage; WVL: leaves of the wild plants at the vegetative stage; CFR: rhizomes of the cultivated plants at the flowering stage; WFR: rhizomes of the wild plants at the flowering stage; CFL: leaves of the cultivated plants at the flowering stage; WFL: leaves of the wild plants at the flowering stage; CWR: rhizomes of the cultivated plants at the withering stage; WWR: rhizomes of the wild plants at the withering stage.

**Figure 3 genes-12-00459-f003:**
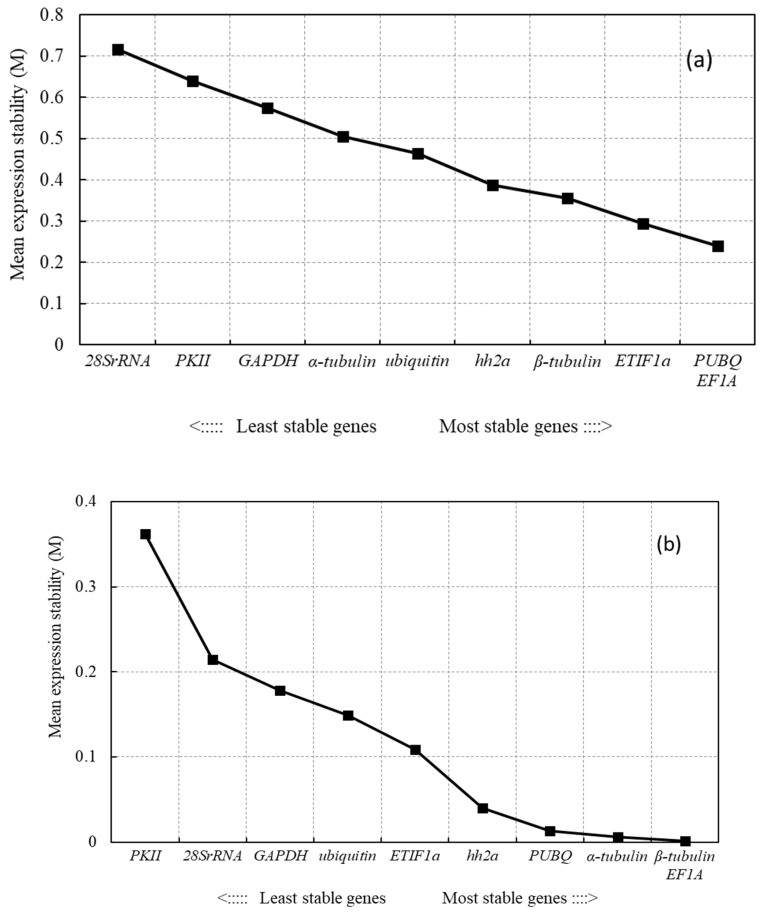
Expression stability and ranking of 10 candidate reference genes calculated by geNorm. The mean expression stability was calculated by a stepwise exclusion of the least stable gene across all samples. The least stable genes are on the leftand the most stable ones on the right. The statistical analysis of the expression stability was performed for all of the tested samples (Total) and separately for each growth stage for the rhizome samples and for the samples of the leaves of the cultivated and wild plants. (**a**) Total: all rhizome and leaf samples of cultivated and wild plants; (**b**) VR: rhizomes at the vegetative stage; (**c**) VL: leaves at the vegetative stage; (**d**) FR: rhizomes at the flowering stage; (**e**) FL: leaves at the flowering stage; (**f**) WR: rhizomes at the withering stage.

**Figure 4 genes-12-00459-f004:**
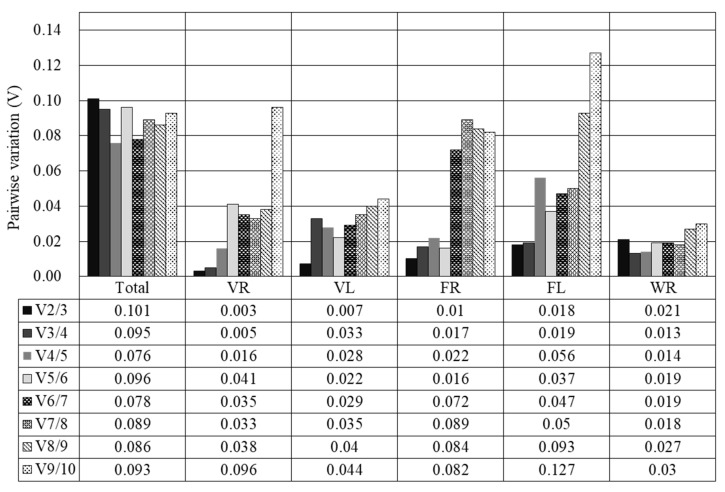
Determination of the optimal number of reference genes required for effective normalization. A pairwise variation (Vn/Vn+1) analysis between the normalization factors (NFn and NFn+1) was performed by the geNorm program to determine the optimal number of reference genes that might be carried out for RT-qPCR data normalization in various sample pools. The statistical analysis of the expression stability was performed for all of the tested samples (Total) and separately for each growth stage for the rhizome samples and for the samples of the leaves of the cultivated and wild plants. Total: all rhizome and leaf samples of the cultivated and wild plants; VR: rhizomes at the vegetative stage; VL: leaves at the vegetative stage; FR: rhizomes at the flowering stage; FL: leaves at the flowering stage; WR: rhizomes at the withering stage.

**Table 1 genes-12-00459-t001:** Primer information and amplification efficiency of 10 candidate reference genes.

Gene	Gene Name	Primer Sequence (5′-3′)	Amplicon Length (bp)	Tm (°C)	Determination Coefficient (R^2^)	E
*β*-*tubulin*	Beta-tubulin	F: GCCTGCTTGAATGTGGAGAATCT	129	57.8	0.980	1.84
		R: CCCTTCACAAATCGCAATCTCAAC		57.9		
*PUBQ*	Polyubiquitin	F: CAAGTGACACCAATGCCCTAAACT	151	57.9	0.975	1.86
		R: GATGGCAGGGTATATTTTCCTACGC		59.6		
*ETIF1a*	Eukaryotic translation initation factor 1A	F: TGTTCTTCGGCATGGCTACT	126	55.4	0.981	2.00
		R: CCACGGCTCTCGTTCATCTAA		57.6		
*ubiquitin*	Ubiquitin	F: CTCATCACCAGCACCTACATC	146	54.9	0.992	2.01
		R: CCGATTCCGCAACCAAGT		54.9		
*PKII*	Pyruvate kinase II	F: GATGATGCTGCGGCTTGAAG	137	57.4	0.983	2.08
		R: CCAACAGACGGATTGGATTATCTC		57.9		
*GAPDH*	Glyceraldehyde-3- phosphate dehydrogenase	F: CCGAGTCCTGGATCTGATT	124	55.6	0.997	2.13
		R: GGGTGCAAACTAGATAACTGG		55.2		
*α-tubulin*	Alpha-tubulin	F: ACATGCGATGTAATGGCAAGAAGC	134	57.9	0.963	1.81
		R: GGTGCTTGTTCTGTTCTCCAGTGA		59.6		
*EF1A*	Elongation Factor 1-α	F: AGGCGGAGAGGCTTATCA	147	54.9	0.984	1.92
		R: GAGGTCTACTAATCTGGACTGGTA		57.9		
*hh2a*	Histone H2A	F: TCAGCTTCAGCTCAAGCACTAACATCAG	145	61.1	0.905	2.00
		R: GGCGTTCCTGTGGTGTAGTTGTATGG		62.7		
*28S rRNA*	28S ribosomal RNA	F: TCTAGTAACGGCGAGTGAAG	156	55.4	0.984	1.94
		R: GGAACTTAGGTCGGTGGTTA		55.4		

E, Amplification efficiency for each gene.

**Table 2 genes-12-00459-t002:** Expression stability of the candidate reference genes calculated by the NormFinder software.

Rank	Total		VR		VL		FR		FL		WR	
	Gene	Stability	Gene	Stability	Gene	Stability	Gene	Stability	Gene	Stability	Gene	Stability
1	*PUBQ*	0.094	*EF1A*	0.002	*PUBQ*	0.002	*PUBQ*	0.004	*PUBQ*	0.004	*EF1A*	0.001
2	*EF1A*	0.167	*PUBQ*	0.003	*ETIF1a*	0.002	*ETIF1a*	0.004	*EF1A*	0.004	*β-tubulin*	0.001
3	*ETIF1a*	0.206	*α-tubulin*	0.004	*EF1A*	0.011	*ubiquitin*	0.031	*ETIF1a*	0.017	*ubiquitin*	0.026
4	*β-tubulin*	0.210	*ETIF1a*	0.006	*α-tubulin*	0.078	*hh2a*	0.106	*β-tubulin*	0.018	*PUBQ*	0.046
5	*hh2a*	0.316	*hh2a*	0.029	*hh2a*	0.091	*PKII*	0.166	*28SrRNA*	0.086	*ETIF1a*	0.055
6	*ubiquitin*	0.344	*β-tubulin*	0.124	*28SrRNA*	0.102	*EF1A*	0.173	*α-tubulin*	0.211	*GAPDH*	0.060
7	*α-tubulin*	0.372	*ubiquitin*	0.162	*PKII*	0.103	*GAPDH*	0.335	*PKII*	0.212	*α-tubulin*	0.083
8	*GAPDH*	0.494	*GAPDH*	0.208	*GAPDH*	0.189	*α-tubulin*	0.441	*ubiquitin*	0.212	*28SrRNA*	0.091
9	*PKII*	0.506	*28SrRNA*	0.297	*ubiquitin*	0.295	*β-tubulin*	0.519	*hh2a*	0.662	*hh2a*	0.188
10	*28SrRNA*	0.635	*PKII*	0.661	*β-tubulin*	0.299	*28SrRNA*	0.562	*GAPDH*	0.877	*PKII*	0.208

The statistical analysis of the expression stability was performed for all of the tested samples (Total) and separately for each growth stage for the rhizome samples and for the samples of the leaves of the cultivated and wild plants. Total: all rhizome and leaf samples of cultivated and wild plants; VR: rhizomes at the vegetative stage; VL: leaves at the vegetative stage; FR: rhizomes at the flowering stage; FL: leaves at the flowering stage; WR: rhizomes at the withering stage.

**Table 3 genes-12-00459-t003:** Expression stability of the candidate reference genes calculated by the BestKeeper software.

Rank	Total	VR	VL	FR	FL	WR
1	*PUBQ*	*PUBQ*	*PUBQ*	*ETIF1a*	*ETIF1a*	*ETIF1a*
CV ± SD	0.14 ± 0.04	0.27 ± 0.07	0.13 ± 0.04	0.07 ± 0.02	0.04 ± 0.01	0.05 ± 0.02
2	*EF1A*	*EF1A*	*hh2a*	*PUBQ*	*PUBQ*	*PUBQ*
CV ± SD	0.76 ± 0.18	0.35 ± 0.08	0.15 ± 0.04	0.10 ± 0.03	0.07 ± 0.02	0.07 ± 0.02
3	*hh2a*	*ETIF1a*	*ETIF1a*	*EF1A*	*β-tubulin*	*EF1A*
CV ± SD	0.96 ± 0.25	0.31 ± 0.09	0.19 ± 0.05	0.12 ± 0.03	0.11 ± 0.03	0.14 ± 0.03
4	*ETIF1a*	*β-tubulin*	*28SrRNA*	*ubiquitin*	*EF1A*	*β-tubulin*
CV ± SD	0.98 ± 0.26	0.31 ± 0.09	0.47 ± 0.14	0.22 ± 0.06	0.22 ± 0.06	0.12 ± 0.04
5	*β-tubulin*	*α-tubulin*	*α-tubulin*	*28SrRNA*	*PKII*	*α-tubulin*
CV ± SD	1.02 ± 0.29	0.42 ± 0.10	0.70 ± 0.15	0.26 ± 0.07	0.78 ± 0.17	0.21 ± 0.05
6	*ubiquitin*	*ubiquitin*	*β-tubulin*	*β-tubulin*	*ubiquitin*	*ubiquitin*
CV ± SD	2.15 ± 0.52	0.47 ± 0.11	0.56 ± 0.16	0.32 ± 0.09	0.79 ± 0.19	0.32 ± 0.08
7	*GAPDH*	*hh2a*	*EF1A*	*hh2a*	*28SrRNA*	*GAPDH*
CV ± SD	2.14 ± 0.55	0.51 ± 0.14	0.70 ± 0.16	1.24 ± 0.33	0.69 ± 0.19	0.38 ± 0.10
8	*PKII*	*GAPDH*	*PKII*	*GAPDH*	*α-tubulin*	*hh2a*
CV ± SD	2.47 ± 0.58	0.53 ± 0.14	0.96 ± 0.23	1.71 ± 0.44	1.37 ± 0.31	0.51 ± 0.14
9	*α-tubulin*	*28SrRNA*	*GAPDH*	*PKII*	*hh2a*	*28SrRNA*
CV ± SD	2.98 ± 0.69	0.91 ± 0.24	1.13 ± 0.29	2.69 ± 0.65	2.19 ± 0.58	0.48 ± 0.14
10	*28SrRNA*	*PKII*	*ubiquitin*	*α-tubulin*	*GAPDH*	*PKII*
CV ± SD	2.56 ± 0.71	2.61 ± 0.63	1.75 ± 0.41	2.88 ± 0.67	2.99 ± 0.78	0.94 ± 0.22

The statistical analysis of the expression stability was performed for all of the tested samples (total) and separately for each growth stage for the rhizome samples and for the samples of the leaves of the cultivated and wild plants. Total: all rhizome and leaf samples of cultivated and wild plants; VR: rhizomes at the vegetative stage; VL: leaves at the vegetative stage; FR: rhizomes at the flowering stage; FL: leaves at the flowering stage; WR: rhizomes at withering stage. Descriptive statistics of 10 candidate genes based on the coefficient of variance (CV) and standard deviation (SD) of their Ct values were determined using the whole data set. Reference genes were identified as the most stable genes with the lowest coefficient of variance and standard deviation (CV ± SD).

## Data Availability

The data presented in this study are available in Appendix A.

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
