# Peer review of "Appropriate Reference Genes for RT-qPCR Normalization in Various Organs of Anemone flaccida Fr. Schmidt at Different Growing Stages"

_genes, 2021, doi:10.3390/genes12030459_

Round 1

Reviewer 1 Report

The manuscript "Appropriate Reference Genes for RT- qPCR Normalization in Various Organs of Anemone flaccida Fr. Schmidt at Different Stages" is well written and the results are clear. The selection of genes by three independent generally used programs was done appropriately. It is a pity that analyses under different stress conditions were not also included. Especially in the case that some stresses could potentially increase the content of medicinally beneficial compounds.

I have a few notes to the manuscript:

I suggest correcting the Title - delete the gap in "RT- qPCR", use italics for the species name, and specify the term "different stages".

The manuscript should also mention the reference Mo et al. 2019. The article used 18S rRNA as the reference gene. You also use references 14,15,16 for the programs but you use other references in the Discussion to explain the principle of the analyses (for example line 282).

In the Introduction, you write that the plants grow vigorously (line 45), but they have also slow growth (line 51). Make it clear. Use also the same abbreviations of genes as in the text (lines 74,75) and move here also the abbreviation description from line 129. The last sentence of the Introduction is not easy to understand (lines 90-93). Please, rewrite it.

Line 108 - correct rhizomes were collected from withering plants on July...

Line 116 - samples were collected in 3 biological replicates

line 118 - 2 g of leaves, 1g of rhizomes

line 128 - correct "in this study: EF1A..."

line 137 - correct gap "5-3 and"

line 138 - Ct values were

Table 1 should be aligned and the name 28S ribosomal RNA corrected

lines 144, 150 - use concentration of primers

line 180,181 - The sentence is not clear.

line 184 - expression level of...

Figure 1 legend - the deviated values are probably not outliers (line 189) but samples of a different stage/organ and the term "outliers" used in the legend is not correct.

I do not agree with the interpretation in Figure 2. The percentage representation can mispresent the data. For example, you write that beta-tubulin is consistent (line 192), but it is not true when you look at Fig.1. Similarly hh2a. It should be better to visualize genes on the x-axis and the different samples in columns on the y-axis. The connected paragraph (lines 190-200) is mispresented according to Fig.2.

line 209 - the tested reference genes by geNorm showed...

line 219 - expressed gene in the rhizomes

line 219 - the sentence Most 3/4 values... should be introduced and add to the context

line 250 - The results obtained by... ; delete the comma after "that"

Table 3 - the description under the table is not correct

line 266 - reference 30 is not about the mentioned species

line 270 - software etc.

line 274 - the sentence should be corrected (it is not clear what "total sample" means)

Table S1 - the abbreviations SR and SL should be changed to VL and VR

Fig. S1 - add the company producing the marker

Fig. S2 - the bands do not correlate with the size of fragments shown in Table 1. The marker should be named.

Fig. S3 - the names alpha and beta tubulin should be on the same line

Reviewer 2 Report

The manuscript I have revised is well written but needs to be improved. After taking into account the comments, it requires re-revision .

Detailes:

  1. The authors assessed the expression of 10 genes for cultivated and wild plants in the vegetative, flowering and withering stages, examining two types of organs: rhizomes and leaves, for the last withering stage only the rhizomes. They marked the samples in the text (line 104-115) with symbols that do not agree with Figure S1, Table S1 - please check and correct it. Whether the vegetative stage is the seedling stage. Please, clarify it and standardize.
  2. Line 129 should be "A. flaccida "instead of" A. Flaccida "
  3. Line 128-131 - gene names and their abbreviations should be identical throughout the manuscript, including tables and figures
  4. Table 1: it should be "Amplicon lenght" instead of "Amplicon Lengh". The column of primer sequences can be expanded to make them more readable
  5. The methodology should clearly and precisely state that the statistical analysis of the expression stability was performed for all the tested samples (total) and separately for each growth stage for the rhizome samples and for the samples of leaves of cultivated and wild plants, which is then reflected in the figures and other descriptions of the figures (Figure 3, Figure 4) as well as in the Table 2, Table 3, Table S1)
  6. Amplicon sizes presented in Table 1 do not agree with the image in Figure S2. Is the gene sequence description correct in Figure S2? For example, 9 is the α-tubulin gene - amplicon size 134 bp (Table 1), and 10 is PUBQ - amplicon size 151 bp (Table 1). On gel, 10 migrates noticeably faster than 9?
  7. It seems to me that the last sentence in the conclusion is not a conclusion from the presented research, but expresses the plans to apply the results of the presented analyzes
  8. References require minor corrections: e.g. line 364, 369, 406-407, 416.
